# Continuous Deep Q-Learning in Optimal Control Problems: Normalized Advantage Functions Analysis

**Anton Plaksin**
IMM UB RAS, Ural Federal University
Yekaterinburg, Russia
`a.r.plaksin@gmail.com`

**Stepan Martyanov**
IMM UB RAS, Ural Federal University
Yekaterinburg, Russia
`martyanovst@gmail.com`

## Abstract

One of the most effective continuous deep reinforcement learning algorithms is normalized advantage functions (NAF). The main idea of NAF consists in the approximation of the Q-function by functions quadratic with respect to the action variable. This idea allows to apply the algorithm to continuous reinforcement learning problems, but on the other hand, it brings up the question of classes of problems in which this approximation is acceptable. The presented paper describes one such class. We consider reinforcement learning problems obtained by the time-discretization of certain optimal control problems. Based on the idea of NAF, we present a new family of quadratic functions and prove its suitable approximation properties. Taking these properties into account, we provide several ways to improve NAF. The experimental results confirm the efficiency of our improvements.

## 1 Introduction

The standard reinforcement learning (RL) [Sutton and Barto, 2018] setup consists of an agent interacting with an environment. At each step of the interaction, the agent determines an action based on its policy and its current state, gets a reward, and makes a transition to the next state. The aim of the agent is to learn the policy that maximizes the sum of rewards.

One of the most widespread approach for solving RL problems is Q-learning [Watkins and Dayan, 1992], which consist in seeking the optimal action-value function (Q-function) as a solution of the Bellman optimality equation. After the learning, the agent can act optimally by the found Q-function. Initially, Q-learning approach was applied for solving RL problems with finite state and action spaces. In this case, Q-function can be represented by a finite table. For the case of a large or continuous state space, Q-learning has later been extended to Deep Q-learning algorithm [Mnih et al., 2015] which allows to seek the approximations of Q-function in the class of neural networks by means of the stochastic gradient descent. Deep Q-learning and its modifications have shown efficiency for a range of challenging tasks [Wang et al., 2016, van Hasselt et al., 2016, Schaul et al., 2016, Hessel et al., 2018]. However, this algorithm can not be directly applied for solving RL problems with continuous action spaces. The reason is that Deep Q-learning involves maximizing of approximate Q-function by the action variable on each step of the learning, which is a complex procedure for continuous action spaces. Among various approaches to overcome this problem [Lillicrap et al., 2016, Haarnoja et al., 2017, Wu et al., 2017, Kalashnikov et al., 2018, Nachum et al., 2018, Fujimoto et al., 2018, Lim et al., 2019, Ryu et al., 2020, Lutter et al., 2021], we focus on the idea of the normalized advantage functions (NAF) algorithm [Gu et al., 2016], which consists in the approximation of Q-function by the function quadratic with respect to the action variable. It allows to calculate the maximum quite fast and precisely and solve some challenging control problems [Gu et al., 2017, Dong et al., 2018, Ikemoto and Ushio, 2021], but on the other hand, it brings up the question of classes of RL problems

36th Conference on Neural Information Processing Systems (NeurIPS 2022).

in which this approximation is acceptable. The presented paper describes one of the possible answers to this question.

Note that the class of LQR problems [Bradtke et al., 1994] has Q-functions quadratic with respect to the action variable. However, this class, being quite special, is not suitable for the description of non-linear controlled processes. In this paper, we study a wider (in some sense) class of RL problems. We consider RL problems which are obtained by time-discretization of finite-horizon optimal control problems [Bardi and Dolcetta, 1997] in dynamical system is described by affine in control ordinary differential equations and the cost functional is quadratic in control. Note that a lot of RL problems with continuous action spaces arise from control problems for mechanical or robotic systems [Lillicrap et al., 2016, Gu et al., 2016, Haarnoja et al., 2017, Gu et al., 2017, Kalashnikov et al., 2018] which are, in fact, described by affine in control ordinary differential equations. Besides, often, the cost functional in a control problem can be determined as quadratic in control. For instance, if the quality index in an initial problem does not depend on control, then one can add a small quadratic term such that, on the one hand, the quality index becomes quadratic, and on the other hand, the sense of the problem does not changed much. Thus, the class of control problems under consideration seems quite general.

For this class, based on the idea of NAF, we present a new family of quadratic functions and prove that, first, this family is sufficiently rich to approximately solve the Bellman optimality equation (Theorem 1), and second, any sufficiently accurate solution, from this family, of the Bellman optimality equation allows to approximately obtain the optimal policy in the corresponding optimal control problem (Theorem 2). Moreover, we prove that it is impossible to get the same results for the original family of functions from Gu et al. [2016] (Theorem 3). Next, we provide several ways to utilize the obtained theoretical knowledge in order to improve NAF. The experimental results confirm the efficiency of our improvements.

Note that, in contrast to LQR problems, we can not say that Q-function is quadratic in our class of problems. Moreover, it is not true in general case. Nonetheless, according to presented statements (Theorem 1,2), Q-function can be approximated by quadratic functions with sufficient order of accuracy to obtain near-optimal policy.

## 2    Background

The standard reinforcement learning (RL) setup consists of an agent interacting with an environment [Sutton and Barto, 2018]. This interaction is described by a Markov Decision Process (MDP), which is a tuple $(\mathcal{S}, \mathcal{U}, \mathcal{P}, \mathcal{R}, \rho_0, \gamma)$, where $\mathcal{S}$ is a state space, $\mathcal{U}$ is an action space, $\mathcal{P}(s'|s, u)$ is a transition distribution, $\mathcal{R}(s, u)$ is a reward function, $\rho_0(s)$ is an initial state distribution, and $\gamma \in [0, 1]$ is a discount factor. The aim of the agent is to learn optimal policy $\mu_*(s)$ which maximizes the value

$$J(\mu) = \mathbb{E}\left[\sum_{i=0}^{\infty} \gamma^i \mathcal{R}(s_i, u_i) \,|\, s_0 \sim \rho_0(s_0),\, u_i = \mu(s_i),\, s_{i+1} \sim \mathcal{P}(s_{i+1}|s_i, u_i), i = 0, 1, 2, \ldots\right].$$

In the general statement of reinforcement learning problems, a policy of the agent can be stochastic, however, within this paper, we assume that the policy is deterministic.

One of the most effective approach for solving RL problems is Q-learning [Watkins and Dayan, 1992], which consist in seeking the optimal action-value function

$$Q_*(s, u) = \sup_{\mu} \mathbb{E}\left[\sum_{i=0}^{\infty} \gamma^i \mathcal{R}(s_i, u_i) \,|\, s_0 = s,\, u_0 = u,\right.$$
$$\left. s_{i+1} \sim \mathcal{P}(s_{i+1}|s_i, u_i),\, u_{i+1} = \mu(s_{i+1}),\, i = 0, 1, 2, \ldots\right]$$

as a solution of the Bellman optimality equation

$$Q_*(s, u) = \mathbb{E}\left[\mathcal{R}(s, u) + \gamma \max_{u' \in \mathcal{U}} Q_*(s', u') \,|\, s' \sim P(s'|s, u)\right].$$

In other words, the agent solves the following minimization problem:

$$\sup_{s \in \mathcal{S}, u \in \mathcal{U}} \left|Q(s, u) - \mathbb{E}\left[\mathcal{R}(s, u) + \gamma \max_{u' \in \mathcal{U}} Q(s', u') \,|\, s' \sim \mathcal{P}(s'|s, u)\right]\right| \to \inf_Q. \tag{1}$$

If the agent knows the function $Q_*(s, a)$, it can act optimally by the greedy policy

$$\mu_*(s) \in \underset{u \in \mathcal{U}}{\operatorname{argmax}} Q_*(s, u).$$

Initially, the Q-learning approach was applied for solving RL problems with finite state and action spaces. In this case, the Q-function is represented by a finite table and problem (1) is finite-dimensional. For the case of a large or continuous state space, Q-learning has later been extended to Deep Q-learning algorithm [Mnih et al., 2015] which allows to seek approximations of Q-function in the class of neural networks $Q(x, u|\theta^Q)$, where $\theta^Q$ is the parameter vector of the neural network. During the learning, the experiences $(s_i, u_i, r_i, s_{i+1})$ are stored in the buffer $\mathcal{D}$ and simultaneously the parameter vector $\theta^Q$ is updated by means of the stochastic gradient descent minimizing the loss function

$$L(\theta^Q) = \mathbb{E}\big[(Q(s, u|\theta^Q) - y)^2 \,|\, (s, u, r, s') \sim U(\mathcal{D})\big], \quad y = r + \gamma \max_{u' \in \mathcal{U}} Q(s', u'|\theta^Q). \quad (2)$$

where $U(\mathcal{D})$ is the uniform distribution on $\mathcal{D}$. Deep Q-learning and its modifications are effective for a range of challenging tasks [Wang et al., 2016, van Hasselt et al., 2016, Schaul et al., 2016, Hessel et al., 2018]. However, note that this algorithm can not be directly applied for solving RL problems with continuous action spaces. The reason is that Deep Q-learning involves the maximizing in (2) on each step of the learning, which is a complex procedure for continuous $\mathcal{U}$. Among various approaches to overcome this problem [Lillicrap et al., 2016, Haarnoja et al., 2017, Kalashnikov et al., 2018, Lim et al., 2019, Ryu et al., 2020, Lutter et al., 2021], we focus on an idea of the normalized advantage functions (NAF) algorithm [Gu et al., 2016]. This idea consists in the approximation of Q-function by the following quadratic with respect to $u$ functions:

$$Q(s, u|\theta^Q) = V(s|\theta^V) + A(s, u|\theta^A),$$
$$A(s, u|\theta^A) = -\frac{1}{2}(u - \mu(s|\theta^\mu))^T P(s|\theta^P)(u - \mu(s|\theta^\mu)), \quad (3)$$

where $V(s|\theta^V)$, $\mu(s|\theta^\mu)$, and $P(s|\theta^P)$ are neural networks with parameters $\theta^V$, $\theta^\mu$, and $\theta^P$, respectively; $P(s|\theta^P)$ is a positive-definite square matrix for each $s$ and $\theta^P$; $\theta^A = \{\theta^\mu, \theta^P\}$ and $\theta^Q = \{\theta^A, \theta^V\}$. Under the condition

$$\mu(s|\theta^\mu) \in U, \quad (4)$$

it allows to get the maximum and argmaximum values directly by values of $V(s|\theta^V)$ and $\mu(s|\theta^\mu)$:

$$\max_{u \in \mathcal{U}} Q(s, u|\theta^Q) = V(s|\theta^V), \quad \underset{u \in \mathcal{U}}{\operatorname{Argmax}} Q(s, u|\theta^Q) = \mu(s|\theta^\mu), \quad (5)$$

but on the other hand, it brings up the question of classes of RL problems in which quadratic approximations is acceptable. Below, we describes one such class.

## 3 Problem statement

In this section, we consider a certain class of optimal control problems and show that discrete approximations of these problems can be formalized as RL problems.

Consider the following optimal control problem: it is required to maximize the functional

$$J(u(\cdot)) = \sigma(x(T)) - \int_0^T \big(q(t, x(t)) + u(t)^T r(t, x(t)) u(t)\big) dt, \quad (6)$$

over all $u(\cdot)$, where $x(\cdot)$ is the solution [Filippov, 1988, §1] of the differential equation

$$\frac{d}{dt} x(t) = f(t, x(t)) + g(t, x(t)) u(t), \quad t \in [0, T], \quad (7)$$

under the initial condition

$$x(0) = z. \quad (8)$$

Here $t$ is the time variable, $T > 0$ is the terminal instant of time, $x(t) \in \mathbb{R}^n$ is the current state vector, $u(t) \in U$ is the current control action vector forming the measurable function $u(\cdot)$, $U \subset \mathbb{R}^m$ is the nonempty compact set, $z \in \mathbb{R}^n$ is the fixed initial state vector, $f(t, x) \in \mathbb{R}^n$, $g(t, x) \in \mathbb{R}^{n \times m}$,

$q(t,x) \in \mathbb{R}$, $r(t,x) \in \mathbb{R}^{m \times m}$, $(t,x) \in [0,T] \times \mathbb{R}^n$ are continuous with respect to $t$ and continuously differentiable with respect to $x$ functions, $r(t,x)$ is the positive-definite matrix for each $(t,x) \in [0,T] \times \mathbb{R}^n$, and $\sigma(x) \in \mathbb{R}$, $x \in \mathbb{R}^n$ is the continuously differentiable function. We assume that there exists a constant $c_{fg} > 0$ such that

$$\|f(t,x) + g(t,x)u\| \leq (1 + \|x\|)c_{fg}, \quad (t,x) \in [0,T] \times \mathbb{R}^n, \quad u \in U. \tag{9}$$

Note that, under these conditions, for each function $u(\cdot)$, there exists a unique solution $x(\cdot)$ of equation (7) under the initial condition (8) [Filippov, 1988, §1].

Define the value function in optimal control problem (6), (7) by

$$V_*(t_*, x_*) = \sup_{u(\cdot)} \left( \sigma(x(T)) - \int_{t_*}^T \left( q(t, x(t)) + u(t)^T r(t, x(t)) u(t) \right) dt \right), \tag{10}$$
$$(t_*, x_*) \in [0,T] \times \mathbb{R}^n,$$

where, for each $u(\cdot)$, $x(\cdot)$ is the solution of equation (7) on the interval $[t_*, T]$ under the initial condition $x(t_*) = x_*$.

Define the sets

$$S = \left\{ (t,x) \in [0,T] \times \mathbb{R}^n \colon \|x\| \leq (1 + \|z\|)e^{c_{fg}t} - 1 \right\}, \quad S(t) = \left\{ x \in \mathbb{R}^n \colon (t,x) \in S \right\}. \tag{11}$$

Let $k \in \mathbb{N}$, $\Delta t_k = T/k$, and $t_i = i\Delta t_k$, $i \in \overline{0,k}$. Consider the corresponding discrete optimal control problem: it is required to maximize the function

$$J_k(u_0, u_1, \ldots u_{k-1}) = \sigma(x_k) - \Delta t_k \sum_{i=0}^{k-1} \left( q(t_i, x_i) + u_i^T r(t_i, x_i) u_i \right), \tag{12}$$

over all $u_i \in U$, $i \in \overline{0, k-1}$, where $(x_0, x_1, \ldots, x_k)$ is defined by

$$x_0 = z, \quad x_{i+1} = x_i + (f(t_i, x_i) + g(t_i, x_i)u_i)\Delta t_k, \quad i \in \overline{0, k-1}. \tag{13}$$

Let us show that problem (12), (13) can be formalized as the RL problem. First, we define the state and actions spaces, the initial state distribution, and the discount factor as follows:

$$\mathcal{S} = \cup_{i=0}^k \left( \{t_i\} \times S(t_i) \right) \cup s_T, \quad \mathcal{U} = U, \quad \rho_0(s_0) = \delta(s_0 = (0, z)), \quad \gamma = 1. \tag{14}$$

Here $s_T$ is some fictional terminal state, $\delta$ is Dirac delta distribution. Next, for every $i \in \overline{0, k-1}$, $x \in S(t_i)$, and $u \in U$, we define the transition distribution and the reward function by

$$\mathcal{P}(s'|s = (t_i, x), u) = \delta(s' = (t_{i+1}, x')), \quad \mathcal{R}(s = (t_i, x), u) = -\left( q(t_i, x) + u^T r(t_i, x)u \right)\Delta t_k, \tag{15}$$

where $x' = x + (f(t_i, x) + g(t_i, x)u)\Delta t_k$. Taking into account (9) and (11), one can prove the inclusion $(t_{i+1}, x') \in \mathcal{S}$. Hence, the transition distribution $\mathcal{P}$ is well-defined. For $i = k$, we set

$$\mathcal{P}(s'|s = (t_k, x), u) = \delta(s' = s_T), \quad \mathcal{R}(s = (t_k, x), u) = \sigma(x), \quad x \in S(t_k), \quad u \in U. \tag{16}$$

In order to make dynamical processes (13) formally infinite, we put

$$\mathcal{P}(s'|s_T, u) = \delta(s' = s_T), \quad \mathcal{R}(s_T, u) = 0, \quad u \in U. \tag{17}$$

Thus, we define MDP which describes the RL problem corresponding to problem (12), (13). Next, we show that such RL problems is suitable for using quadratic approximations of the Q-function.

## 4 Quadratic approximations of the Q-function

Denote by $\mathbb{Q}$ the family of functions $Q$ such that

$$Q(t, x, u) = V(t, x) + A(t, x, u), \quad (t, x, u) \in [0, T) \times \mathbb{R}^n \times U, \tag{18}$$

where

$$A(t, x, u) = -(u - \tilde{\mu}(t, x))^T P(t, x)(u - \tilde{\mu}(t, x))$$
$$+ (\mu(t, x) - \tilde{\mu}(t, x))^T P(t, x)(\mu(t, x) - \tilde{\mu}(t, x)) \tag{19}$$
$$\mu(t, x) \in \operatorname*{argmin}_{u' \in U} (u' - \tilde{\mu}(t, x))^T P(t, x)(u' - \tilde{\mu}(t, x)).$$

Here $V(t, x)$, $\tilde{\mu}(t, x)$, and $P(t, x)$ are continuous functions; $P(t, x)$ is a positive-definite square matrix for each $(t, x) \in [0, T] \times \mathbb{R}^n$.

Note that, functions $Q$ from the family $\mathbb{Q}$ satisfy the equalities

$$\max_{u \in U} Q(t, x, u) = V(t, x), \quad \underset{u \in U}{\mathrm{Argmax}}\, Q(t, x, u) = \mu(t, x), \qquad (20)$$

as well as (see (5)) quadratic functions from family (3). However, these function families are different. The difference is that we do not assume the inclusion $\tilde{\mu}(t, x) \in U$ as opposed to assumption (4). In other words, the vertex of a quadratic form (3) belongs to the set $U$, while this is not required in the function family $\mathbb{Q}$. Thus, $\mathbb{Q}$ is a wider family of quadratic functions.

The theorems below establish a connection between the optimal control problem (6)–(8) and minimization problems (1) for MDP (14)–(17).

**Theorem 1.** Let the value function $V_*(t, x)$ be continuously differentiable. Then, for every $\varepsilon > 0$, there exists $k_* > 0$ such that, for every $k \geq k_*$, the function $Q \in \mathbb{Q}$ defined by (18) and (19) where

$$V(t, x) = V_*(t, x), \quad P(t, x) = r(t, x)\Delta t_k, \quad \tilde{\mu}(t, x) = \frac{1}{2} r^{-1}(t, x) g^T(t, x) \nabla_x V_*(t, x) \quad (21)$$

satisfies the inequality

$$\left| Q(t_i, x, u) + \big(q(t_i, x) + u^T r(t_i, x) u\big)\Delta t_k - \max_{u' \in U} Q(t_{i+1}, x', u') \right| \leq \varepsilon \Delta t_k,$$

$$x' = x + \big(f(t_i, x) + g(t_i, x)u\big)\Delta t_k, \quad u \in U, \quad x \in S(t_i), \quad i \in \overline{0, k-1}, \qquad (22)$$

where we assume $Q(t_k, x', u') = \sigma(x')$.

**Theorem 2.** Let the value function $V_*(t, x)$ be continuously differentiable. Let $\varepsilon > 0$ and $k_* > 0$ be defined according to Theorem 1. Take $k \geq k_*$ and suppose that a function $Q \in \mathbb{Q}$ satisfies inequality (22). Then the following estimate holds:

$$J_k(u_0, u_1, \ldots u_{k-1}) \geq \sup_{u(\cdot)} J(u(\cdot)) - 3T\varepsilon, \qquad (23)$$

where the function $J_k(u_0, u_1, \ldots u_{k-1})$ is defined by (12) with $u_i = \mu(t_i, x_i)$, $i \in \overline{0, k-1}$ and the function $\mu(t, x)$ is defined by $Q$ according to (18) and (19).

Thus, Theorem 1 shows that the function family $\mathbb{Q}$ is sufficiently rich to contain approximate solutions of problem (1) with a predetermined accuracy and Theorem 2 establishes that all such approximate solutions contained in $\mathbb{Q}$ allow us to get the policy, which approximately provides the optimal result in optimal control problem (6)–(8).

Now, let us consider the original family of functions from Gu et al. [2016]. Denote by $\mathbb{Q}_{NAF}$ the family of functions $Q$ such that

$$Q(t, x, u) = V(t, x) - (u - \mu(t, x))^T P(t, x)(u - \mu(t, x)), \quad (t, x, u) \in [0, T] \times \mathbb{R}^n \times U, \quad (24)$$

where $V(t, x)$, $\mu(t, x)$, and $P(t, x)$ are continuous functions; $P(t, x)$ is a positive-definite square matrix for each $(t, x) \in [0, T] \times \mathbb{R}^n$; $\mu(t, x) \in U$ for each $(t, x) \in [0, T] \times \mathbb{R}^n$. The theorem below establishes that if we take the family $\mathbb{Q}_{NAF}$ instant of $\mathbb{Q}$, then Theorem 1 can not be proved even in the simplest cases of optimal control problem (6)–(8).

**Theorem 3.** Let $n = m = 1$, $T = 1$, $U = [-1, 1]$, $f(t, x) = q(t, x) = 0$, $g(t, x) = r(t, x) = 1$, $\sigma(x) = -x^2$, and $z = 2$. Then, for every $k \geq 4$ and $Q \in \mathbb{Q}_{NAF}$, there exist $i \in \overline{0, k-1}$, $x \in S(t_i)$, and $u \in U$ such that

$$\left| Q(t_i, x, u) + u^2 \Delta t_k - \max_{u' \in U} Q(t_{i+1}, x', u') \right| > \Delta t_k/8, \quad x' = x + u \Delta t_k,$$

where we assume $Q(t_k, x', u') = -(x')^2$.

Proofs of the theorems are given in the Appendix A.

Note that, in the particular case $U = \mathbb{R}^m$, the family $\mathbb{Q}$ coincides with family $\mathbb{Q}_{NAF}$. However, within this paper, $U$ is required to be bounded, which is significantly for the presented theorems. Indeed, unboundedness $U$ implies unboundedness the set $S$ (see (11)) which make it impossible to obtain inequality (22) uniformly on this set. Thus, these theorems are not applicable to any unbounded $U$ and, in particular, $U = \mathbb{R}^m$.

Table 1: Parameters in the examples of optimal control problems

| Name | $n$ | $m$ | $T$ | $U$ | $f(t,x)$ | $g(t,x)$ | $q(t,x)$ | $r(t,x)$ | $\sigma$ | $z$ |
|---|---|---|---|---|---|---|---|---|---|---|
| Van der Pol oscillator | 2 | 1 | 11 | $[-1,1]$ | $\begin{pmatrix} x_2 \\ (1-x_1^2)x_2 \end{pmatrix}$ | $\begin{pmatrix} 0 \\ -x_1 \end{pmatrix}$ | 0 | 0.05 | $\sigma_1(x)$ | $\begin{pmatrix} 1 \\ 0 \end{pmatrix}$ |
| Pendulum | 2 | 1 | 5 | $[-2,2]$ | $\begin{pmatrix} x_2 \\ 14.7\sin(x_1) \end{pmatrix}$ | $\begin{pmatrix} 0 \\ 3 \end{pmatrix}$ | 0 | 0.01 | $\sigma_2(x)$ | $\begin{pmatrix} \pi \\ 0 \end{pmatrix}$ |
| Dubins car | 3 | 1 | $2\pi$ | $[-0.5,1]$ | $\begin{pmatrix} \cos(x_3) \\ \sin(x_3) \\ 0 \end{pmatrix}$ | $\begin{pmatrix} 0 \\ 0 \\ 1 \end{pmatrix}$ | 0 | 0.05 | $\sigma_3(x)$ | $\begin{pmatrix} 0 \\ 0 \\ 0 \end{pmatrix}$ |
| A target problem | 6 | 2 | 10 | $[-1,1]^2$ | $\begin{pmatrix} 0 \\ 0 \\ x_5 \\ x_6 \\ x_1-x_3 \\ x_2-x_4 \end{pmatrix}$ | $\begin{pmatrix} 1 & 0 \\ 0 & 1 \\ 0 & 0 \\ 0 & 0 \\ 0 & 0 \\ 0 & 0 \end{pmatrix}$ | 0 | 0.001 | $\sigma_4(x)$ | $\begin{pmatrix} 0 \\ 0 \\ 0 \\ 0 \\ 0 \\ 0 \end{pmatrix}$ |

## 5 Experiments

We consider four examples of optimal control problems (6)–(8) described in Table 1, where

$$\sigma_1(x) = -x_1^2 - x_2^2, \quad \sigma_2(x) = -|x_1| - 0.1|x_2|, \quad \sigma_3(x) = -|x_1 - 4| - |x_2| - |x_3 - 0.75\pi|$$

$$\sigma_3(x) = -x_1^2 - x_2^2 - (x_3 - 2)^2 - (x_4 - 2)^2$$

Van der Pol oscillator is a famous model of a non-conservative oscillator with non-linear damping. The aim of the control is to stabilize the oscillator at the terminal time.

Pendulum is a traditional problem for testing control algorithms. The aim of the control is the stabilization of the pendulum in the top position at the terminal time.

Dubins car is a quite famous model which describes a motion of the point particle moving at a constant speed on the plane. The problem is to find a control providing the closeness of the motion with a target point at the terminal time.

A target problem is an optimal control problem presented in Munos [2006]. The dynamic system describes a hand holding a spring to which is attached a mass. It is required to control the hand such that the mass achieve the target point at the terminal time.

### 5.1 Bounded NAF

First, we modify NAF algorithm, proposed in Gu et al. [2016], based on the function family $\mathbb{Q}$. Note that, the considered examples have $U = [\alpha, \beta]^m$. Denote $\text{clip}(\nu) = \max\{\alpha, \min\{\beta, \nu\}\}$, $\nu \in \mathbb{R}^m$, $\alpha, \beta \in \mathbb{R}^m$. Then, according to (18) and (19), we can use the following approximation of the Q-function, within NAF algorithm:

$$Q(t, x, u|\theta^Q) = V(t, x|\theta^V) + A(t, x, u|\theta^A),$$

$$A(t, x, u|\theta^A) = -\big(u - \tilde{\mu}(t, x|\theta^{\tilde{\mu}})\big)^T P(t, x|\theta^P)\big(u - \tilde{\mu}(t, x|\theta^{\tilde{\mu}})\big)$$

$$+ \big(\text{clip}(\tilde{\mu}(t, x|\theta^{\tilde{\mu}})) - \tilde{\mu}(t, x|\theta^{\tilde{\mu}})\big)^T P(t, x|\theta^P)\big(\text{clip}(\tilde{\mu}(t, x|\theta^{\tilde{\mu}})) - \tilde{\mu}(t, x|\theta^{\tilde{\mu}})\big),$$

where $V(t, x|\theta^V)$, $\tilde{\mu}(t, x|\theta^{\tilde{\mu}})$, and $P(t, x|\theta^P)$ are neural networks with parameters $\theta^V$, $\theta^{\tilde{\mu}}$, and $\theta^P$, respectively; $P(t, x|\theta^P)$ is a positive-definite square matrix for each $(t, x)$ and $\theta^P$; $\theta^A = \{\theta^{\tilde{\mu}}, \theta^P\}$ and $\theta^Q = \{\theta^A, \theta^V\}$. To be short, we call this algorithm Bounded NAF (BNAF), because it is essential for our modification of NAF that the set $U$ is bounded.

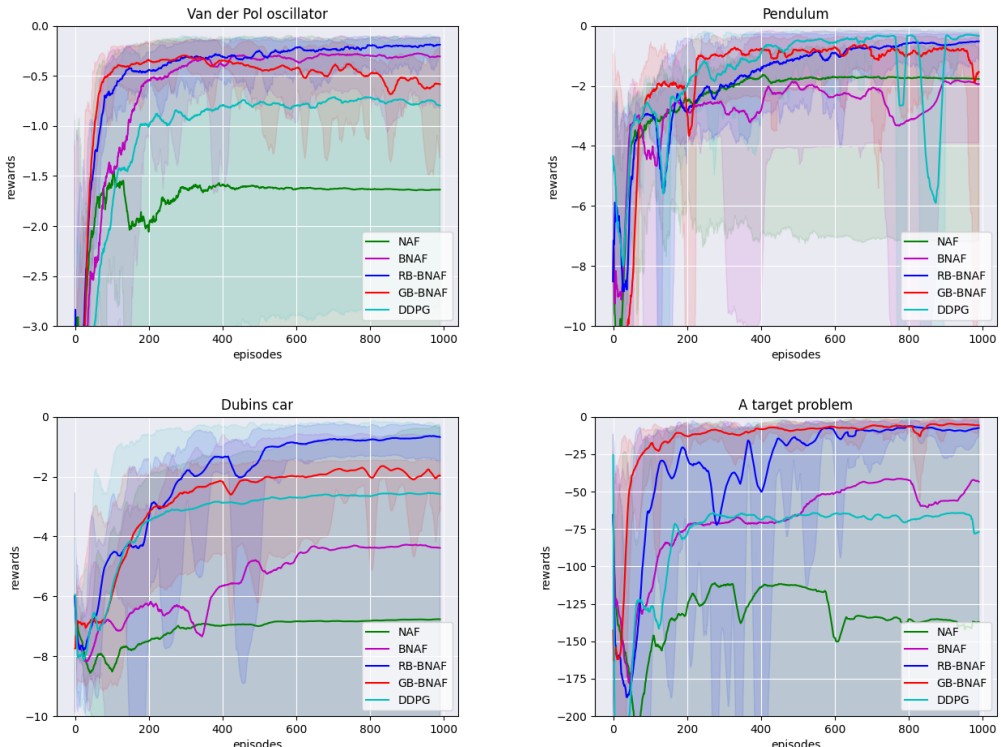

Figure 1: Performance of NAF, BNAF, RB-BNAF, GB-BNAF, DDPG algorithms averaged over 10 seeds. The curves are averaged for 20 episode for better visualization.

## 5.2 Reward-based BNAF

In practice in many cases, the function $r(t, x)$ is known. For example, if a researcher who formalizes a control problem has the right to determine an appropriate cost functional. Then, based on the second equality in (21), the function $r(t, x)\Delta t_k$ can be used instead of the neural network $P(t, x|\theta^P)$ to reduce the number of learning parameters. This variant of BNAF is called Reward-based BNAF (RB-BNAF).

## 5.3 Gradient-based BNAF

One can also imagine the situation in which, taking into account mechanical laws and experimental measurements, the function $g(t, x)$ also turns out to be known for the researcher. Then, according to the third equality in (21), the function

$$\tilde{\mu}(t, x|\theta^V) = \frac{1}{2}r^{-1}(t, x)g^T(t, x)\nabla_x V(t, x|\theta^V)$$

can be used instead of the neural network $\tilde{\mu}(t, x|\theta^{\tilde{\mu}})$, which also reduces the number of learning parameters. This variant of BNAF is called Gradient-based BNAF (GB-BNAF).

## 5.4 Experimental results

We use the same learning parameters of every our tasks. We apply neural networks with two layers of 256 and 128 rectified linear units (ReLU) and learn their used ADAM with the learning rate $lr = 1e^{-3}$. We use batch size $n_{bs} = 128$ and smoothing parameter $\tau = 1e^{-2}$. Also we take $\Delta t = 0.1$. All calculations were performed on a personal computer in a standard way.

We compare NAF, BNAF, RB-BNAF, GB-BNAF algorithms, and DDPG algorithm presented in [Lillicrap et al., 2016]. We utilize the hyperbolic tangent on the output layer of the neural network

corresponding to the policy $\mu(s|\theta^\mu)$ in NAF and DDPG algorithms in order to provide inclusion (4). Figure 1 shows learning curves of the algorithms for the considered examples. One can note that RB-BNAF algorithm has a good performance in all examples. GB-BNAF is slightly inferior to RB-BNAF everywhere except A target problem. DDPG and BNAF have excellent results only one example out of four. NAF algorithm shows not the worst performance only in Pendulum example.

Thus, taking the presented experiments into account, we can give the following general recommendations. For solving optimal control problems (6)–(8), it is rational to use GB-BNAF and especially RB-BNAF algorithms along with NAF and DDPG algorithms. There is a high probability that they will show better results and more stable learning.

Let us also emphasize that the presented algorithms can be successfully applied to problems without the dependence of the cost functional (6) on $u(t)$. In this case, the quadratic term with small $r(t, x(t))$ can be added in the initial problem. Then, on the one hand, smallness of this term does not result in poor performance of the algorithms (see pendulum and a target problem examples) and on the other hand the obtained results are close to the solution of the initial problem.

# 6    Related works

Many different ways to apply reinforcement learning approaches for solving optimal control problems are investigated [Baird, 1994, Doya, 1995, 2000, Munos, 2006, Tallec et al., 2019, Kim and Yang, 2020, Lutter et al., 2021]. Among them, a time-discretization is perhaps the most obvious and widely used tool [Lillicrap et al., 2016, Gu et al., 2016, Haarnoja et al., 2017, Gu et al., 2017, Kalashnikov et al., 2018]. From the theoretical point of view, it is known [Bardi and Dolcetta, 1997, p.388] that solutions of time-discrete optimal control problems converge to the solution of the initial problem as the discretization step tends to zero. In the present paper, we also use the time-discretization and study approximating solutions of Bellman equations and the corresponding greedy policies depending on the discretization step (see Theorems 1 and 2). Other studies of dependencies on the discretization step of reinforcement learning methods can be found in Munos [2006], Tallec et al. [2019].

We focus on the idea from Gu et al. [2016] to expand Q-learning algorithm to optimal control and reinforcement learning problems with continuous actions. Other approaches for solving such problems are investigated in Lillicrap et al. [2016], Haarnoja et al. [2017], Kalashnikov et al. [2018], Lim et al. [2019], Ryu et al. [2020], Lutter et al. [2021]. The paper Lutter et al. [2021] seems the closest to the presented paper. In Lutter et al. [2021], the similar optimal control problem and feedback control policy are considered, but the proposed algorithms require knowledge of the right-hand side of differential equations in contrast to algorithms suggested in presented paper.

Let us also note the family of functions (18), (19) is included, in some sense, to families of Q-functions considered in Wang et al. [2016], Tallec et al. [2019]. It seems to be expected because more general classes of problems are considered in these papers. Nevertheless, the proposed algorithms and results are very different from presented in this paper.

# 7    Conclusion

We consider reinforcement learning problems which are obtained by time-discretization of finite-horizon optimal control problems described by affine in control ordinary differential equations and quadratic in control cost functionals. Based on the idea of NAF algorithm Gu et al. [2016], we present a new family of quadratic functions and prove that, first, this family is sufficiently rich to approximately solve the Bellman optimality equation (Theorem 1), and second, any sufficiently accurate solution, from this family, of the Bellman optimality equation allows to approximately obtain the optimal policy (Theorem 2). Moreover, we establish that it is impossible to get the same results for the function family from Gu et al. [2016] (Theorem 3). Next, we utilize the obtained theoretical results to propose new algorithms improving NAF. The experimental results confirm their efficiency.

## Acknowledgments and Disclosure of Funding

The work was performed as part of research conducted in the Ural Mathematical Center with the financial support of the Ministry of Science and Higher Education of the Russian Federation (Agreement number 075-02-2022-874).

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
