# OpenReview forum: "Continuous Deep Q-Learning in Optimal Control Problems: Normalized Advantage Functions Analysis"
_NeurIPS.cc/2022/Conference — NeurIPS 2022 Accept_

### Official Review · Reviewer_mYi8 · 2022-07-06

**Rating:** 6
**Confidence:** 4
**Soundness:** 3 good
**Presentation:** 2 fair
**Contribution:** 2 fair

**Summary:**

The paper provides analysis of a method for solving optimal control problems based on approximating the value function by means of deep neural networks that results in a quadratic parametrization of the value function in the control effort, thus allowing for an efficient solution of the optimization step needed for deep Q learning. Several theorems are proved to identify and analyze the class of optimal control problems this method is applicable to. Several variants of the algorithms are investigated empirically on four test problems, in comparison with one well known method for continuous Q learning, DDPG.

**Questions:**

Is DDPG really the current state of the art to be used as a baseline? Shouldn't PPO and/or SAC also be included in empirical comparisons?

On page 2, line47, by "quality index", do you mean the cost (respectively, reward) function?

A minor typo:
P.8 L.199: "greedy politics" -> "greedy policies"

**Limitations:**

I think the authors have adequately addressed the limitations of the proposed methods - in fact, a major objective of the paper is to analyze and understand which classes of optimal control problems these methods are applicable (and possibly, limited) to. I do not see any negative societal impact of this work.

**Strengths And Weaknesses:**

The main contribution of the paper is probably the empirical investigation of the proposed method that illustrates that two variants of the algorithm, RB-BNAF and GB-BNAF, have significantly higher performance than DDPG on all test problems, and as such, could be good candidates for solution methods for optimal control problems of a suitable class. Analysis demonstrates that this class included non-linear control-affine dynamical systems with cost functions quadratic in the control effort. These methods could be a good addition to the toolbox of practitioners. Both methods require at least partial knowledge of the system being controller (the reward function and the control-affine term in the dynamics), so these methods cannot be considered proper reinforcement learning methods, but as optimal control methods, they appear to be valid and quite competitive.

However, because the basic NAF algorithm is already known in the literature, this paper's contribution is relatively minor. The class of optimal control problems the methods are applicable to is important in practice, but not that universal. The authors are arguing that by adding a quadratic term to the cost function, many control problems can be made to be of the kind they are considering, but this is a somewhat contrived way of claiming the generality of the method.

---

> ### Author Response · Authors · 2022-08-02
> **Response**
>
> First of all, thank you for your review and positive attitude to the proposed algorithms.
>
> In response to your concerns, indeed, it is quite difficult to reliably judge how common a certain class of optimal control problems is. One can rely only on own experience and knowledge of the problems usually appeared in various research. Based on this, it seems to us that the quadraticity of a cost functional is not the most common condition for optimal control problems. However, we would like to emphasize that even if a cost functional does not have quadratic term (which seems to consider more often) this is not an obstacle to apply the proposed algorithms. One can always add a small quadratic term and thus get into the considered class. Moreover, taking into account the results of our experiments, it can be seen that the smallness of this term does not lead to poor performance of the algorithms. This is what we tried to say when discussing generality. In order to clarify this point, we have put additional comments at the end of Section 5.4.
>
> Speaking about the comparison of the presented algorithms with others, we would like to emphasize that we posed the main goal of our paper as the investigation and development of the NAF algorithm idea, that is quadratic approximations of Q-functions. Therefore, we compere the proposed algorithms with the standard NAF and standard DDPG preceding it. The idea of a quadratic approximations of Q-functions (or A-functions) can apparently be used in other reinforcement learning algorithms such as SAC and maybe even PPO. Then, the comparison with these algorithms would seem more natural. On the whole, this is a good idea for further research, but it seems so general that it is hardly to be implemented within a single study. In this sense, the presented paper can be considered as the next research in this direction after [Gu at el., 2016].
>
> Thanks again for your feedback. We hope that our response has helped to clear up some doubts and our edits to the paper have made it better. If this is the case in your opinion, then we respectfully ask that you consider increasing your score. If you have any more comments, questions or remarks, we would be glad to discuss them.

---

> > ### Comment · Reviewer_mYi8 · 2022-08-08
> > **Generality of the method**
> >
> > Thank you for your response to my questions. It is certainly possible to put any optimal control problem into this class by adding a small quadratic term, but what I was trying to say was that this is a pretty contrived way of demonstrating generality. What if the cost functional has a dominant term that is not quadratic in the control effort, but something else, for example cubic. If you add a quadratic component, you will achieve a locally quadratic optimization surface, and you will be able to apply the analytical method of computing the optimum, but wouldn't this optimum be only local? if yes, what useful purpose would this small quadratic term serve?

---

> > > ### Author Response · Authors · 2022-08-08
> > > **Response**
> > >
> > > Thank you for your questions.
> > >
> > > Above, we spoke exclusively about the case when a cost functional has only the term $\sigma(x(T))$. In our opinion, such problems are quite often encountered, and, in this case, a small quadratic term helps us place a problem in the class discussed in the paper. However, if a cost functional has other nonlinear terms (for example, a cubic term), then, you are right, the quadratic addition does not help. For such problems, quadratic approximations of q-functions do not fit. It seems to us that such problems are not often encountered, therefore we are talking about a certain generality of our class. That is, we mean the generality not from the mathematical point of view, but from the practical point of view.

---

### Official Review · Reviewer_e5w3 · 2022-07-07

**Rating:** 5
**Confidence:** 4
**Soundness:** 3 good
**Presentation:** 2 fair
**Contribution:** 2 fair

**Summary:**

Q-learning is a popular approach to reinforcement learning that is difficult to extend to continuous action spaces. Normalized advantage functions (NAFs) address this issue by assuming that the Q function approximation is quadratic with respect to the controls. This enables us to determine the policy by efficiently computing the argmax with respect to controls of our approximated Q function. While it seems to work in practice, which systems this approximation is suitable for is an open question. To address this concern, the paper analyzes the case where the dynamics are control-affine, the cost is quadratic in the controls, and the problem is formulated in continuous time. First, the authors map the continuous-time optimal control problem to a discrete-time reinforcement learning problem. Next, they propose a slight variation of the NAF approximation family and show how to set the terms given knowledge of the known optimal control problem to bound the Q-function approximation error. They then present how this translates into a bound on the performance. Additionally, they show that these bounds do not hold if we use the original NAF approximation via a counterexample. Finally, the authors evaluate a couple variants of their new approximation family on a set of benchmarks and compare their performance to both the baseline NAF implementation and a policy gradient method.

**Questions:**

*  If the controls are unconstrained, would Theorems 1 and 2 hold for the original NAF approximation or is there something more that would prevent this?
* Are the observed benefits of BNAF and its variants really due to this new approximation family? Would I observe similar benefits if I used the modifications to NAF I discussed above?

**Limitations:**

The authors do discuss the limitations of their work in the introduction. Specifically, they mention that the analysis in this paper only holds for a specific class of problems. Moreover, the Q function approximation they propose is not the true form, even in the more limited class of problems they consider. However, they are able to bound the approximation error and performance, which significantly adds to the paper.

**Strengths And Weaknesses:**

The question of which control problems the NAF approximation is acceptable to is a relevant and important question. Moreover, the approximation and performance bounds presented in this paper are applicable to a useful class of problems that apply to a wide variety of real-world robotics tasks. The authors also present a number of variants of their Q-function approximations which can incorporate knowledge about the reward and dynamics if known. The experimental results illustrate the benefits of this extended approximation family and the variants in terms of the overall reward achieved on a number of benchmarks. Additionally, the paper is fairly well organized and explains most of the relevant background material.

However, the addition of control constraints to the problems considered seems to be changing the question the paper originally asked. Specifically, there's the question of which problems the NAF approximation is suitable for, and then there's the question of how to adapt NAF in the case of control constraints. Both are useful questions to ask, but they need to be disentangled somehow. For instance, it is unclear if the proposed extended approximation family would be important in the case of unconstrained controls. The main difficulty in applying Theorem 1 and 2 to the original NAF approximation appears to be that we cannot set the policy directly to the optimal as in (21) due to the presence of these constraints. This is what motivates the introduction of the new family in (18, 19). However, in the case where the controls are unconstrained, it seems like the original NAF approximation should be sufficient. While the theorems presented in this paper are interesting and useful regardless, the authors should elaborate on this point.

Moreover, even if the controls are constrained, it is hard to tell from the experiments if the benefits observed really come from this new family or just from incorporating additional information into the Q function. For instance, in the case of box constraints on the controls, one could easily set $\mu$ in the original NAF family such that the neural network output is also clipped or passed through a scaled sigmoid. Similarly, for GB-BNAF, one could also set $\mu$ to be $\frac{1}{2} r^{-1}(t,x) g^T(t, x) \nabla_x V(t, x | \theta^V)$ but clipped or passed through a scaled sigmoid. It would greatly strengthen the paper to disambiguate the source of benefits by performing some ablations.

Finally, there are a number of grammatical errors throughout the paper that need to be addressed. A sentence or two describing other approaches besides NAF to tackle continuous actions in Q learning would add to the paper. It would also be nice for the authors to include a conclusion section instead of just ending on the related work.

---

> ### Author Response · Authors · 2022-08-02
> **Response**
>
> Thank you for your feedback. We are glad that you find the paper well organized and the addressed issues relevant and important.
>
> Thank you also for drawing attention to rather subtle issues regarding the boundedness of the set $U$. First of all, let us mention that, indeed, in the original paper [Gu at el., 2016], the set $U$ is formally arbitrary. However, apparently, in all experiments of the paper, this set is bounded and the authors apply hyperbolic tangent for the output layer of the neural network corresponding to the policy in order to the output belongs $U$. Therefore, regarding the boundedness of the set $U$ in the experiments, the presented paper and [Gu at el., 2016] are the same.
>
> The case of unbounded $U$ seems also interesting, but require additional investigations. For example, in the case $U=\mathbb R^n$, the function family $Q$ is simplified and coincides with the family from [Gu at el., 2016] (if we use the linear activation function on the output layer instant of hyperbolic tangent). But on the other hand, the case of unbounded $U$ prevents to obtain results similar to Theorem 1, 2. Indeed, unboundedness $U$ implies unboundedness the set $S$ (see (11)) which make it impossible to obtain inequality (22) uniformly on this set. Perhaps, there are particular cases in which it is possible. For instance, it could be LQR problems. However, this class of the control problems quite specific and require special approach. In our paper, on the contrary, we aim to find a quite general class of optimal control problems (with nonlinear dynamical systems) in which it would be possible to justifiably use quadratic approximations of q-functions.
>
> In our experiments, as well as in [Gu at el., 2016], we use NAF algorithm with the hyperbolic tangent activation function on the output layer to satisfy the control constraints. Moreover, as we were convinced within our additional experiments not included in the paper, the hyperbolic tangent is indeed the best activation output function for NAF. For BNAF algorithms, we use the clip function on the output layer, since, firstly, it's more correct in terms of formula (19) and, secondly, algorithms with the clip function demonstrate better performance (compared with the hyperbolic tangent and linear functions). Thus, for all algorithms, we have chosen the output layer that demonstrates the best performance, and therefore it seems that the difference in their results is caused precisely with different ways to approximate q-functions.
>
> Thanks again for your feedback. We added some comments regarding to the bounded issue of the set $U$ in lines 156-160 and regarding to output activation functions in lines 199-201. We have also included more references (line 31) to papers dedicated to algorithms for continuous action spaces and have added the conclusion (lines 236-245).
>
> We hope that our response has helped to clear up some doubts and our edits to the paper have made it better. If this is the case in your opinion, then we respectfully ask that you consider increasing your score. If you have any more comments, questions or remarks, we would be glad to discuss them.

---

> > ### Comment · Reviewer_e5w3 · 2022-08-08
> > **Response to author rebuttal**
> >
> > Thank you for your clarification with respect to my concerns on the boundedness of the controls. It is clear that this assumption is necessary to prove the theorems in the paper. Moreover, you also consider this when implementing NAF, which invalidates my concerns about the source of benefits. I am also glad to see that you clarified these points in the paper and added a conclusion section. I have adjusted my score accordingly.

---

> > > ### Author Response · Authors · 2022-08-09
> > > **Response**
> > >
> > > Thank you very much. We are glad that we could clarify these details for you and make the paper better.

---

### Official Review · Reviewer_RkHa · 2022-07-11

**Rating:** 5
**Confidence:** 3
**Soundness:** 3 good
**Presentation:** 3 good
**Contribution:** 2 fair

**Summary:**

This data identifies a class of (discretized) optimal control problems for which a quadratic (with respect to actions) approximation to the problem's Q-function is accurate and gives rise to near-optimal policies.  The proposed quadratic parameterization of the Q-function is similar to, but slightly more general than, that proposed by the original Normalized Advantage Functions paper, and this additional generality is necessary.  The authors test several implementations of this approximation on control problems and find it to be effective.

**Questions:**

(1) How does the performance of BNAF (+ variants) compare to more state-of-the-art RL algorithms (e.g. SAC) for continuous action spaces, and to what extent is BNAF compatible with various implementation details that improve performance in these baselines?

(2) How gracefully does BNAF handle violations of the assumptions in equation (6)?

**Limitations:**

The authors are clear about the limitations of BNAF and related algorithms, unambiguously defining a set of problems to which these approaches are applicable.



**Strengths And Weaknesses:**

This paper provides a useful theoretical contribution by characterizing a set of RL tasks for which NAF-style Q-function approximations are suitable.  The explanations and derivations are fairly clear and the experiments are easy to follow.

My principal concern with the paper is that its main contribution, while useful, is rather limited in scope.  It is unclear to me whether NAF-style approaches are competitive with the state of the art in continuous action-space RL problems.  The authors do compare against DDPG, which is nice, but not against stronger baselines like SAC.  The relevance of this paper's theoretical contribution to the community is ultimately somewhat dependent on the performance of NAF-style algorithms relative to alternatives.

I think the paper has another major shortcoming independent of the above concern, which is that there is no analysis of problems that *do not* lend themselves to a quadratic Q-function approximation.  Obviously the space of all MDPs is too large to categorize in a single paper, but the authors could help clarify the applicability of NAF by analyzing which aspects of the problem formulation in (6) are essential to the paper's theoretical and empirical results.  For instance, how does the suitability of the NAF approximation, and its actual performance, scale as the dependence of cost on u(t) becomes increasingly non-quadratic?

Finally, two comments on presentation: (1) I think the motivation for the equation (19), and the three implementations in section 5, could beimproved.  As I understand it, the deviations from the original NAF algorithm are made to account for the possibility that the optimal action in a given state may not belong to the set of actions to the agent.  This point could be made clearer in the text.  (2) The reward-based BNAF and gradient-based BNAF algorithms are introduced rather abruptly, without much motivation.  Under what conditions are we likely to know r(t, x) or g(t, x)?  When should we intuitively expect (without seeing the empirical results) these algorithms to outperform vanilla BNAF?

---

> ### Author Response · Authors · 2022-08-02
> **Response**
>
> First of all, thank you for your comments and questions.
>
> We posed the main goal of our paper as the investigation and development of the NAF algorithm idea, that is quadratic approximations of Q-functions.  Therefore, we compere the proposed algorithms with the standard NAF and standard DDPG preceding it. SAC appears to be a more advanced version of DDPG in which an entropy term and two Q-network are used to improve its performance. In response to your question, it seems that these tools could also be exploited for the NAF algorithm and the algorithms proposed in the paper. In this case, the comparison with SAC would be more correct. To improve performance of the algorithms, we can also use many others tools known in literature (such as prioritize experience replay, various architectures of neural networks, various noises and so on). This is a good direction for future research, however, within this paper, we tried to focus on more basic things.
>
> Speaking of (6), if your question is about (we apologize if we do not understand it quite right) the dependency of theoretical and experimental results on decrease $r(t,x)$ and can the results be applied to the case $r(t,x) = 0$, then response is as follows. For the theoretical results, a positive definiteness of $r(t,x)$ is essential, but $r(t,x)$ can be arbitrarily small. However, in practice, we can applied the algorithms even to the case of $r(t,x)=0$. To do this, the quadratic term with small $r(t,x)$ can be added to the initial problem. Then, on the one hand, smallness of this term does not result in poor performance of the algorithms (see pendulum and a target problem examples in which $r(t,x) = 0.01$ and $r(t,x)=0.001$, respectively) and on the other hand the obtained results are close to the solution of the initial problem. Moreover, we can estimate the difference. For example, in Pendulum problem, it is less than $T \times r(t,x) \times  u_{max}=0.1$. We put additional comments about this at the end of Section 5.4.
>
> We also put some additional comments about (19) in lines 130-132. Speaking more informally, the difference between function family (3) and the function family $Q$ and is that the vertex of a quadratic form (3) belongs to the set $U$, while this is not required in $\mathbb Q$. Thus, $\mathbb Q$ is a wider family of quadratic functions.
>
> Finally, based on your remarks on the deduction of RB-BNAF and GB-BNAF algorithms, we add some motivation in lines 182-185 and 188-191. Namely, function $r(t,x)$ can be known, for example, if a researcher who formalizes a control problem has the right to determine an appropriate cost functional. This situation seems to happen quite often. In this case, based on the second equality in (21), the function $r(t,x) \Delta t_k$ can be used instead of the neural network $P(t,x|\theta^P)$ to reduce the number of learning parameters. One can also imagine the situation in which, taking into account mechanical laws and experimental measurements, the function $g(t,x)$ also turns out to be known for the researcher. Then, according to the third equality in (21), the function defined under the line 190 can be used instead of the neural network $\tilde{\mu}(t,x|\theta^{\tilde{\mu}})$, which also reduces the number of learning parameters.
>
> Thanks again for your feedback. We hope that our response has helped to clear up some doubts and our edits to the paper have made it better. If this is the case in your opinion, then we respectfully ask that you consider increasing your score. If you have any more comments, questions or remarks, we would be glad to discuss them.

---

> > ### Comment · Reviewer_RkHa · 2022-08-07
> > **Follow-up question**
> >
> > Thank you for the thoughtful responses to my comments.  Before considering an updated evaluation of the paper, I want to clarify one of my questions since I seem not to have phrased it clearly enough in the original review.  As I understand it, not all RL problems are suitable for the quadratic Q-function approximation used in this paper.  The paper considers a restricted class of RL problems (described by a discretized version of equations 6-8) for which the approximation is accurate.  What I am interested in is how violating the assumptions of equations (6) and (7) affects the goodness of approximation (the result of Theorem 1).  For instance, how important is the assumption that $\frac{d}{dt} x(t)$ is linear in u(t)?  And how important is it that the cost function in (6) is quadratic in u(t) rather than having, say, a linear component, or a higher-order component?  Which of these assumptions is most likely to be violated in realistic control problems (say, e.g., standard RL benchmarks from OpenAI gym?)

---

> > > ### Author Response · Authors · 2022-08-08
> > > **Response**
> > >
> > > Thank you for your questions.
> > >
> > > If the cost functional does not have a quadratic term or it has a quadratic term together with an additional non-linear with respect to $u$ term or a dynamical system depends on $u$ non-linear, then we can not prove Theorem 1. In other words, these conditions are essential for Theorem 1.
> > >
> > > Talking about the problems from OpenAI gym, it should be emphasized that they are not problems with a fixed time interval and therefore do not fit into the class under consideration. Nevertheless, if the dynamical system is linear with respect to $u$, then we can try to formalize the meaning of the problem within our class. This is exactly what we did in the Pendulum example. Pendulum from OpenAI Gym (https://github.com/openai/gym/blob/master/gym/envs/classic_control/pendulum.py) is an infinite-horizon problem in which the maximization of cost functional implies stopping the pendulum at the top position with minimal effort (i.e., there is a quadratic term). We formalized this problem as a problem on the finite time interval $[0,5]$ with the cost functional of form (6), the maximization of which leads to a control that provides the top position of the pendulum at the time $T=5$ with minimal effort. Thus, it seems that the linearity with respect to $u$ of a dynamic system is a fairly common case (for example, all classic control problems from OpenAI gym), while, in order to make a cost functional as (6), we may need to change the formalization of a problem insignificantly (adding a small quadratic term) or significantly (as we did with the Pendulum example).

---

### Meta-Review · Area_Chair_gnCx · 2022-08-30

**Recommendation:** Accept
**Confidence:** Less certain

**Metareview:**

I am happy to recommend accepting this paper.

I would argue that the stated contribution of this paper - an analysis of when NAF is a good approximation - is somewhat minor.  But the analysis in the paper has a nice additional benefit in that it becomes possible to include domain-specific knowledge (e.g., on the reward function, which I agree is often known) in a straightforward and effective way into the algorithm.  Furthermore, the paper is fairly well executed.

I would encourage the authors to have another careful look at all the comments by all the reviewers, which could make the paper better.  I would particularly agree that an analysis (empirical or theoretical or both) of what happens when the assumptions are violated would make the paper better.

**Award:**

No

---

### Decision · Program_Chairs · 2022-09-14

Accept